# Antagonist Activation Measurement in Triceps Surae Using High-Density and Bipolar Surface EMG in Chronic Hemiparesis

**DOI:** 10.3390/s24123701

**Published:** 2024-06-07

**Authors:** Mouna Ghédira, Taian Martins Vieira, Giacinto Luigi Cerone, Marco Gazzoni, Jean-Michel Gracies, Emilie Hutin

**Affiliations:** 1Laboratoire Analyse et Restauration du Mouvement (ARM), Hôpitaux Universitaires Henri Mondor, Assistance Publique-Hôpitaux de Paris (AP-HP), 94000 Créteil, France; ghedira.mouna@gmail.com (M.G.); jean-michel.gracies@aphp.fr (J.-M.G.); 2Laboratory for Engineering of the Neuromuscular System, Politecnico di Torino, 10129 Turin, Italy; taian.martins@polito.it (T.M.V.); glc.cerone@gmail.com (G.L.C.); marco.gazzoni@polito.it (M.G.)

**Keywords:** hemiparesis, high-density EMG, soleus, gastrocnemius, coefficient of antagonist activation

## Abstract

After a stroke, antagonist muscle activation during agonist command impedes movement. This study compared measurements of antagonist muscle activation using surface bipolar EMG in the gastrocnemius medialis (GM) and high-density (HD) EMG in the GM and soleus (SO) during isometric submaximal and maximal dorsiflexion efforts, with knee flexed and extended, in 12 subjects with chronic hemiparesis. The coefficients of antagonist activation (CAN) of GM and SO were calculated according to the ratio of the RMS amplitude during dorsiflexion effort to the maximal agonist effort for the same muscle. Bipolar CAN (BipCAN) was compared to CAN from channel-specific (CsCAN) and overall (OvCAN) normalizations of HD-EMG. The location of the CAN centroid was explored in GM, and CAN was compared between the medial and lateral portions of SO. Between-EMG system differences in GM were observed in maximal efforts only, between BipCAN and CsCAN with lower values in BipCAN (*p* < 0.001), and between BipCAN and OvCAN with lower values in OvCAN (*p* < 0.05). The CAN centroid is located mid-height and medially in GM, while the CAN was similar in medial and lateral SO. In chronic hemiparesis, the estimates of GM hyperactivity differ between bipolar and HD-EMGs, with channel-specific and overall normalizations yielding, respectively, higher and lower CAN values than bipolar EMG. HD-EMG would be the way to develop personalized rehabilitation programs based on individual antagonist activations.

## 1. Introduction

In chronic hemiparesis (over six months after the causal brain lesion), walking impairment is associated with a disturbed voluntary movement, especially characterized by inappropriate antagonist muscle activation and insufficient agonist recruitment during voluntary command directed to the agonist [1,2,3,4,5]. Abnormal antagonist recruitment has been shown to be, at least in part, a misdirection of the supraspinal drive triggered by a volitional command to agonist muscles and proportional to its intensity [6,7]. Stretching of the triceps surae intensifies the untimely, antagonistic recruitment of plantar flexors, as bipolar electromyograms (EMG) detected a greater amplitude of plantar flexor coactivation during isometric contractions with the knee extended than with the knee flexed in a seated position [6]. Consistent observations of abnormal coactivations were recorded from the gastrocnemius medialis and from the soleus during the knee re-extension of the late swing phase of gait [7], supporting the relevance of monitoring the degree of antagonist muscle excitation in controlled and ecological conditions.

Muscle activation can be recorded from the skin surface using classic bipolar EMG or with a grid of smaller electrodes (high-density EMG, HD-EMG). The former is thought to provide a single, overall indication of the degree and timing of muscle activity [8,9,10]; this ability may, however, be undermined due to the limited sensitivity provided via a single, recording location [11]. HD-EMG, conversely, may provide a more accurate representation of changes in the degree of muscle excitation, sampling from various small regions within the muscle [12,13]. This high-density approach has been explored in the literature in healthy and stroke subjects [14,15,16]. These validated methods [17,18,19,20] rely on the capacity to sample action potentials from motor units whose fibers reside in various locations within the target muscle. This capacity could be hampered due to the presence of few detection points [13,21,22,23], the anatomy of the underlying tissues [24], or a low signal-to-noise ratio [25]. Regional differences in muscle activation must, therefore, be accounted for if inferences on the whole muscle level are to be made or if focal chemical treatments, such as neuromuscular blocking agents, are considered [26,27]. This issue has been shown to be relevant when attempting to assess antagonist activation. With a grid of electrodes, Vinti and colleagues (2018) [28] assessed the excitation of ankle antagonist and agonist muscles while healthy subjects isometrically contracted their muscles. These authors observed that bipolar EMGs agreed with HD-EMG in measuring tibialis anterior and soleus excitation but not with the gastrocnemius medialis [28]. It is unknown whether that observation extends to persons with hemiparesis. Another open issue to quantify antagonist activation is the method of normalization for surface EMG signals [29], particularly for the HD-EMG. There is the possibility for each signal from the grid of electrodes to perform normalization on a channel-specific basis, i.e., a strict, *focal* approach, in which the denominator of the ratio is a value obtained using exactly the same recording field as the numerator, thus from exactly the same geographical and anatomical environment, or to use the value obtained when extending the recording field across the whole grid as a denominator (the *overall* approach).

The present study first aimed to test whether bipolar EMG and HD-EMG when using channel-specific or overall normalization provided comparable estimates of antagonist GM activations in persons with chronic stroke. Second, the study aimed to assess the spatial distribution of antagonist activations in the GM from an HD-EMG measurement and the possible differences between the medial and lateral portions of the soleus.

## 2. Materials and Methods

### 2.1. Participants

The present prospective, descriptive cohort study was conducted in compliance with the Declaration of Helsinki (2008), the Good Clinical Practice guidelines, the approval of a local ethical committee, and regulatory requirements (registration number: 2015/61NI, Comité de Protection des Personnes Ile-de-France IV). The study was registered in the French national agency for the safety of medicines and health products (identifier: ID-CC2017) and ClinicalTrials.gov (identifier: NCT06099132). All participants were recruited from a neurorehabilitation department at a university hospital. The inclusion criteria were an age ≥ 18 years, hemiparesis due to a non-evolutive central nervous system lesion, a time since the lesion ≥ 6 months, the ability to walk 10 m barefoot without assistance, the cognitive ability to understand verbal instructions for the test according to the investigator’s judgment, and the absence of botulinum toxin injections within the last three months prior to enrollment. All subjects provided written consent before study enrollment. The present manuscript conforms to the Strengthening the Reporting of Observational Studies in Epidemiology guidelines [30].

### 2.2. Experimental Protocol

After being instructed about the experimental procedures, subjects were asked to sit comfortably on an isokinetic dynamometer CON-TREX MG (Con-Trex AG, Dübendorf, Switzerland). The paretic leg was secured to the machine crank, with the lateral malleolus aligned coaxially with the machine axis of rotation. The ankle joint was secured at 90° between the fifth metatarsal bone and the fibula, keeping the hip flexed at 70°. In two knee positions, flexed at 90° and extended at 0°, the subjects performed ≥three-second maximal isometric voluntary contraction (MVC) and then submaximal contractions at 30% MVCs. Visual feedback on ankle torque was provided to the subjects on a screen monitor placed roughly two meters in front of them (Figure 1). During the submaximal trials, the subjects were asked to maintain the torque within the 30 ± 5% MVC target range. For each contraction level and each knee joint angle, one dorsiflexion effort and one plantar flexion effort were performed. The trials were applied in a random order, with 5-min intervals in between. The subjects were encouraged to maintain torque within the target range.

### 2.3. Positioning of Surface Electrodes and EMG Recording

Monopolar HD-EMGs were recorded using arrays of electrodes from GM and SO. As shown in Figure 2, arrays were positioned at specific sites identified with the assistance of ultrasound imaging, following the procedures reported by Vinti and colleagues (2018) [28]. Briefly, after placing the reference electrode (1-cm diameter) on the lateral malleolus, monopolar EMGs were recorded with the following:-A matrix of 64 electrodes (13 × 5 with a missing corner electrode and an 8-mm inter-electrode distance) placed over the GM. The grid was centered halfway between the medial and lateral boundaries of the muscle, with the top row being placed about 2 cm distally to the popliteal fossa.-Two arrays of 8 electrodes (a 5-mm inter-electrode distance) were placed over the medial and lateral portions of SO. Each array was rotated ~45° outward with respect to the leg axis and centered 30 mm distally to the GM myotendinous junctions.

The participants’ skin was cleansed with abrasive paste (Nuprep Skin Prup Gel, weaver and company) before electrode placement. Raw, monopolar EMGs were amplified with 150 *v*/*v* and then sampled at 2048 Hz using a 16-bit A/D converter. EMGs and torque data were sampled synchronously (Quattrocento, OT Bioelettronica, Turin, Italy).

Bipolar EMGs of the GM were simulated from the potentials obtained using the grids of electrodes. Following the same procedure described in a previous work [28], bipolar EMGs were obtained by simulating a pair of monopolar electrodes. Specifically, to mimic the detection provided via electrodes as large as those used in prior studies (2 cm^2^; [6]), we averaged monopolar EMGs collected via consecutive surface electrodes. The center of the simulated, monopolar electrodes was positioned according to the recommendations for the positioning of bipolar electrodes [8,9]. Bipolar EMGs were then computed by taking the difference between the two monopolar EMGs, providing one signal for the GM.

### 2.4. EMG Analysis

First, single-differential EMGs were computed by taking the algebraic difference between monopolar EMGs detected via consecutive rows of electrodes. The root mean square (RMS) amplitude of EMGs was calculated over the whole recording for the submaximal contractions. For maximal efforts, the RMS amplitude was calculated over 500-ms epochs centered at the peak rectified EMG value [6]. In both cases, RMS values were computed for each single-differential EMG separately for each contraction level, knee position, contraction direction, and muscle. The coefficient of antagonist activation (CAN, [7]) was computed.

-
*CAN measurement from bipolar EMG.*


The CAN was calculated according to the ratio of the RMS amplitude during the antagonist effort and the maximal agonist effort for the same muscle. The CAN was, therefore, computed for the GM in each condition (BipCAN_GM_).

-
*CAN measurement from HD-EMG.*


Two different approaches to HD-EMG normalization were applied:(i)The ratio of the RMS value was obtained for a given muscle, acting as an antagonist, and the RMS value obtained for the same muscle while it acted as an agonist at a maximal contraction level on a channel-specific basis—a procedure named channel-specific normalization. Using this approach, we computed the channel-specific CAN (CsCAN) in the GM and SO (CsCAN_GM_ and CsCAN_SO_).(ii)The ratio between RMS values was obtained using the maximal RMS value across EMGs in the grid in the denominator—a procedure named overall normalization. Using this approach, we computed the overall CAN (OvCAN) for the two muscles (OvCAN_GM_ and OvCAN_SO_).

Data processing was conducted using customized Matlab scripts (version 7.1, The MathWork, Inc., Natick, MA, USA).

-
*Spatial distribution of antagonist activation from HD-EMG.*


To describe the spatial distribution of CAN coefficients within GM, we used a fully automated procedure to segment CAN images [31] separately for each condition tested. The number of segmented channels was then normalized with respect to the total number of channels in the grid located over the superficial aponeurosis, thus providing an indication of the relative size of the antagonist activation muscle volume. The average location with the greatest CAN values from the grid of GM electrodes was assessed. Specifically, the centroid of CAN values was quantified as the weighted average of the segmented channels along both the proximal–distal axis (i.e., along the rows of electrodes; Figure 3) and the medial–lateral axis (i.e., along the columns of electrodes) and separately for the two normalization procedures. The centroid coordinates were expected to indicate the location where greater antagonist activation was represented within the grid of GM electrodes. To account for anatomical differences between subjects, the centroid coordinates were normalized in relation to the length of the superficial aponeurosis, defined from where propagation could no longer be observed in the surface EMGs and the popliteal fossa [31]. Along the proximal–distal axis in the GM, zero indicates the most proximal position of the centroid and one the most distal position. Along the medial–lateral axis in the GM, zero indicates the most medial position of the centroid and one the most lateral position (Figure 3). In addition, regionalized antagonist activation in the SO was assessed according to the average CAN in the medial and lateral portions of the muscle.

### 2.5. Statistics

The BipCAN was compared with the CsCAN and the OvCAN in the GM, at two knee positions and two effort levels, using four-factor (effort, position, EMG detection system, and normalization method) ANOVAs after verifying the Gaussian distribution (Kolmogorov–Smirnov test; *p* > 0.150 in all cases) and Bonferroni corrections to test the effect of the EMG detection system, knee position, effort level, and their interactions. The location of the OvCAN and CsCAN centroids in GM was compared between the two knee positions and the two effort levels using post hoc comparisons. In addition, OvCAN and CsCAN in SO were compared between the medial and lateral portions of the muscle with two knee positions and two effort levels using ANOVAs. All statistical analyses were conducted with the Statistica (version 7.0, StatSoft, Inc., Tulsa, OK, USA) software package. A *p*-value of 5% was used for statistical significance.

## 3. Results

### 3.1. Participants

Twelve individuals with chronic hemiparesis participated in the study (Table 1), and 58% of the subjects could not maintain the target torque in 100% MVC for 3 s during the dorsiflexion tasks. All data were still retained for analysis since the experience of the experimenters confirmed that these subjects were indeed attempting to maximally dorsiflex their ankles, as requested.

### 3.2. CAN Measurement in GM Using Bipolar EMG vs. HD-EMG

Clear between-EMG system differences were observed for BipCAN_GM_ and CsCAN_GM_ (main effects, *p* < 0.001), with BipCAN_GM_ yielding lower values than CsCAN_GM_ independent from the effort level and knee position (Table 2, Figure 4A). Between-EMG system differences were also observed between BipCAN_GM_ and OvCAN_GM_, with an interaction effect between the EMG system and effort level (*p* < 0.05), with BipCAN_GM_ yielding higher values than OvCAN_GM_ during 100% dorsiflexion MVC: post hoc Bonferroni, 100% MVC; BipCAN_GM_, 0.51 ± 0.38; OvCAN_GM_, 0.39 ± 0.23; *p* < 0.001, Table 2, Figure 4B). Across all EMG detection systems, normalization approaches, and knee positions, the CAN level increased with effort intensity (*p* < 0.001; Table 2; Figure 4C).

### 3.3. CAN Distribution in GM and SO

Differences were observed when comparing the coordinates of the CAN centroid in the GM calculated from CsCAN and OvCAN (main effect, *p* < 0.01), with an interaction effect between the normalization procedure and knee position (normalization*knee position, *p* < 0.05; Table 3). The CAN centroid estimated from CsCAN was located more proximally compared with OvCAN in the knee-extended condition (*p* < 0.001). The comparisons to other conditions and along the medial–lateral axis of the GM yielded non-significant results.

The CsCAN_SO_ and OvCAN_SO_ values remained similar in the medial and lateral portions of the SO for all conditions tested (main and interaction effects, ns; Table 3).

## 4. Discussion

This study revealed that bipolar and HD-EMGs from the GM provided different estimates of the degree of antagonist activation. This difference depended on the normalization method, with channel-specific and overall normalizations providing respectively higher and lower CAN values when compared to bipolar EMG. Using HD-EMG, the CAN centroid in GM is located mid-height and medially in the muscle, while no differences were found between medial and lateral antagonist activation in the soleus (SO).

### 4.1. Why Considering Two Normalization Procedures in HD-EMG Process?

The issue of normalization has long been a concern in the surface EMG literature [29], with different procedures being recommended for different circumstances [32]. Here, we expand this issue for grids of electrodes, computing antagonist activation coefficients on a channel-specific basis (*focal approach*) or using a denominator across the grid of electrodes (*overall approach*). The motivation for a *focal approach* stems from the possibility that various electrodes in the grid cover various anatomical regions of the same muscle and that a numerator is best compared with a denominator obtained from the exact same recording area. Notwithstanding the documented, anatomical differences between healthy and paretic muscles, such as fascicle shortening and muscle thinning in the plantar flexors [33,34,35,36], whether proximal-to-distal differences in muscle architecture or subcutaneous thickness exist remains an open issue. While this focal approach may contend with anatomical inhomogeneities within the muscle, it is expectedly highly sensitive to local changes in muscle excitation. If, at a maximal contraction level, subjects did not equally activate the whole muscle volume, spuriously high CAN values might have emerged in the grid: indeed, with the focal, channel-specific approach, we observed that some channels provided CAN values greater than 100% (Figure 3). CAN values > 100% in individual cases are actually a classic feature when measuring coefficients of antagonist activation, even when using the bipolar approach [6,7]. One might explain these >100% coefficients of antagonist activation by some motor neurons being more readily fired via the descending coactivation pathways—which are potentially synergistic with afferent activity from potential tension applied within the coactivating muscle, depending on its position—than via the descending voluntary agonist command [7], in addition to the potential measurement variability inherent to EMG. Given the thorough procedure we used in placing the grid, selecting only electrodes covering the GM-superficial aponeurosis (cf. Figures 3 and 6 in Vieira & Botter, 2021 [11]), the observation of CAN values greater than 100% for the focal approach is in agreement with the view that regional differences in GM excitation may, indeed, take place between plantar- and dorsiflexion contractions (Figure 5). Conversely, the *overall* approach using an extended area for the denominator calculation would be expected to bias CAN estimates should there be any electrode in the grid not covering GM or covering GM regions with different architectures. Even though we did not use any imaging technique to control for anatomical inhomogeneities within GM, we are unaware of any documentation of variable pennation angles, subcutaneous thicknesses, local temperatures, local sudation, skin impedance, or other parameters capable of modifying the manner in which muscle action potentials may be transmitted to surface electrodes across the GM areas where we sampled electrical activity. In that context, we elected to present the findings for both normalization approaches here.

### 4.2. Limitations and Differences in Estimations of Antagonist Activation: Detection Modes and Normalization Procedures

Quantified estimations of antagonist activation from both bipolar and HD-EMG in chronic hemiparesis are scarce in the literature. We compared the two detection modes, and the results corroborate those obtained for healthy subjects via the overall normalization procedure [28], providing lower values for coefficients of antagonist activation than with a bipolar EMG. Indeed, BipCAN_GM_ was higher than OvCAN_GM_, especially with the knee extended at maximal effort, and this was also the case for TA in the knee-flexed position. The most plausible explanation for this difference seems to be the regional distribution of muscle excitation in the paretic muscles. Although this possibility would appear undermined due to evidence showing the enlargement of motor unit territories following reinnervation after central lesions [37,38,39], for all subjects tested, we observed that CAN values were not distributed equally across the grid of electrodes (Figure 3): the number of segmented channels ranged from 10 to 30 for the channel-specific normalization and from 9 to 28 for the overall normalization across all trials and all participants. These figures indicate that the size of the excited GM regions was much smaller than half of the muscle volume covered by the grid of electrodes. Possibly, motor unit enlargement is sufficiently small for sub-volumes of the GM to be excited independently, as documented in different circumstances for different lower limb muscles [22,40]. Indirect estimates, indeed, suggest that muscle fibers of motor units in the paretic GM are scattered over a 30% wider region when compared to those in the healthy GM, still spanning a relatively small fraction of the whole muscle length [39]. Given that multiple electrodes cover a large muscle region, local inhomogeneities in antagonist activation are likely accounted for with HD-EMGs, though not with bipolar EMGs. Conceivably, the pickup volume of a pair of bipolar electrodes, placed as considered here (cf. *Methods* and [8,9]), may not convey a sufficiently representative muscle volume. Consequently, bipolar EMGs may provide a biased estimation of the degree of antagonist activation in paretic calf muscles.

An alternative, opposite issue to the spatially limited view provided via surface electrodes is crosstalk [41]. In some cases, the fibers of the target muscle, as well as those of other, perhaps antagonist muscles [42], may be included within the pickup volume of the surface electrodes. Even though we acknowledge the possibility of crosstalk, we see two reasons why this is unlikely to explain the results reported here. First, since crosstalk originates from a distant muscle, due to the conduction volume effect [43,44,45], any crosstalk would be expected to manifest with equal amplitudes across the various electrodes in the grid: it would not explain the regional differences in excitation observed here or, thus, the differences between bipolar and HD-EMGs. Second, the inter-electrode distances considered here are sufficiently small to minimize crosstalk: for a 10-mm inter-electrode distance, we observed crosstalk from the SO to the GM of less than 5% [11]. With our high-density system of electrodes, we believe we have contended with crosstalk while highlighting differences in the topographical activity distribution in the GMs of healthy subjects [28]. In hemiparetic subjects, for the two knee positions and for the two effort levels, antagonist activation with channel-specific normalization provided centroid values confined, on average, from 35% to 45% of the muscle’s longitudinal axis. This range increased to 35–75% when the overall normalization procedure was considered. For the knee-extended position in particular, centroids obtained with the overall normalization were located 20% more distally than those obtained with the channel-specific normalization procedure. These local differences in centroid values, together with the relatively small number of segmented channels and the differences in CAN values between detection modes, indeed suggest that gastrocnemius agonist and antagonist activities are located differently in subjects with hemiparesis as well.

The spatial, differential representation of GM activity during plantar- and dorsiflexion may explain the differences in CAN values obtained with the *channel-specific and overall* normalization procedures. As pointed out above in the previous section, a lower EMG amplitude during an agonist contraction would result in greater CAN values for a given, fixed degree of antagonist activation. Given that the region where the greatest EMGs were detected differed between agonist and antagonist activation, normalization on a channel-specific basis resulted in higher CAN values. While we see the channel-specific normalization as a robust procedure for contending with anatomical inhomogeneities in the target muscle, we might suggest that its use for computing CAN be limited to situations in which the distribution of the EMG amplitude across the grid changes to negligible extents between agonist and antagonist contractions. An alternative, though unexplored, solution to this issue would be using supramaximal nerve stimulation to determine the maximal EMG amplitude that could be potentially detected via each electrode in the grid. Overall, CAN estimates in paretic muscles are highly dependent on the detection system and normalization procedure employed.

### 4.3. Antagonist Activation Equally Estimated in Medial and Lateral Portions of Soleus

Antagonist activation with HD-EMG was equally estimated between the medial and lateral SO portions. For the two muscle portions sampled with the linear arrays, no regional differences in excitation were observed either, in opposition to the GM muscle. Two crucial issues should be considered here. First, the absence of differences suggests a homogeneous antagonist activity at least for the portions studied, as previously observed in healthy subjects [28]. Despite the fact that the soleus is constituted of fibers with a similar physiological profile [46], the muscle has an in-depth pinnate architecture [13]. Because of this in-depth architecture, while features such as action potential propagations may be observed for electrodes positioned at the medial region of the soleus [47], spatial changes in EMG amplitude may be appreciated for EMGs collected in correspondence with the Achilles tendon. The two arrays we used were, therefore, expected to be sensitive to any local changes in SO excitation. The similar CAN values obtained for SO, for the two normalization procedures and all conditions tested, suggest that no significant local changes in excitation took place. A second issue to consider, though, is that the results we obtained may not be generalizable to the whole SO muscle. Only the very distal fraction of the SO is available for the recording of surface EMGs, with its two proximal thirds being covered by the GM. While no medio-lateral differences in CAN values for the SO were apparent, these estimations of the degree of antagonist activation should be interpreted with caution and may not apply to the whole SO volume.

## 5. Conclusions

Both HD-EMG and bipolar EMG have their own strengths and applications, but the choice between the two techniques depends on the specific research or clinical objectives, the level of spatial resolution required, and the available resources and expertise. Furthermore, the non-equivalence between the two systems seems to depend on the normalization procedure selected. Bipolar EMG, with its larger distance between the two electrodes, typically captures higher-amplitude signals due to the larger voltage differences across the muscle region of interest, and this approach is commonly used in clinical practice for the routine assessment of muscle function, such as evaluating muscle disorders or diagnosing neuromuscular conditions. Clinicians should be aware that the amount of antagonist activation caused via bipolar EMG may be under- or overestimated when compared with that obtained using HD-EMG, and this warrants further exploration to refine the clinical responses needed for clinicians when exploring exact treatment effects. HD-EMG could be used to develop personalized rehabilitation programs based on individual antagonist activations and neuromuscular capacities. Indeed, using HD-EMG could be a way to enhance patient education about the role and benefits of their active rehabilitation programs, improving adherence to and engagement with their therapeutic protocols. Moreover, the usefulness and efficacy of botulinum toxin injections used to treat antagonist activation has been suggested to depend on the toxin diffusion from the injection site. HD-EMG could be an efficient method to develop a functional model for predicting the number and location of muscle fibers paralyzed due to the action of the botulinum toxin.

## Figures and Tables

**Figure 1 sensors-24-03701-f001:**
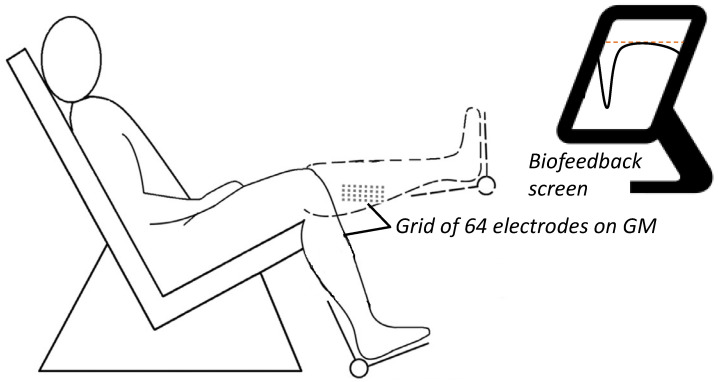
Experimental setup. A grid of 64 electrodes monitoring EMG activity from the gastrocnemius medialis (GM) across two knee positions: flexed at 90° and extended at 0° at maximal isometric voluntary contraction (MVC) and then submaximal contractions at 30% MVC. An isometric brace with a strain gauge measured the dorsiflexor and plantar flexor torque exerted around the ankle. Visual feedback on ankle torque was provided on a screen monitor placed roughly two meters in front of the subjects.

**Figure 2 sensors-24-03701-f002:**
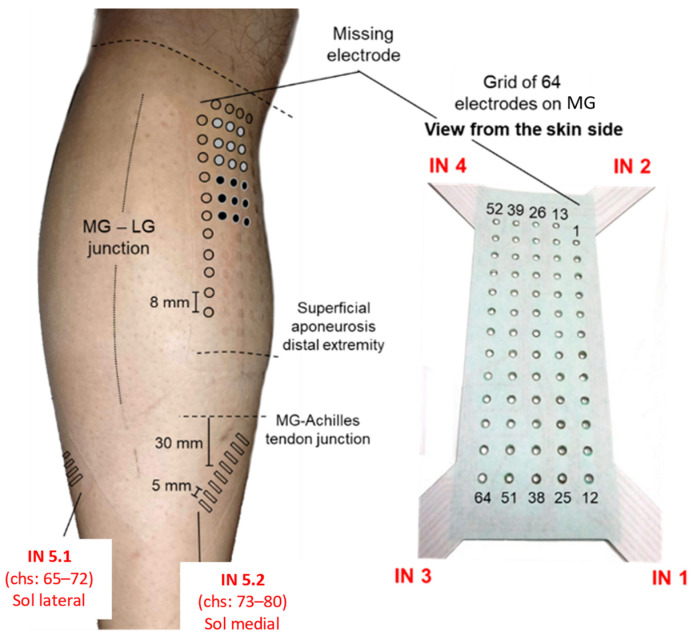
EMG electrodes’ positioning. The relative position of the different systems of electrodes used to sample surface EMGs. The red text, comprising acronyms from IN 1 to IN 6, provides a reference for opening and reading the data stored via the acquisition software. Schematic illustration of the relative positions of surface electrodes on the skin regions covering the medial gastrocnemius (MG) and soleus (SO) muscles. The specific position of surface electrodes is represented by their prints on the skin: circles (MG) and rectangles (SO) were drawn on the picture to emphasize the prints left on the skin upon the electrodes’ removal. The two monopolar electrodes (two groups of 3 × 3 electrodes), considered to simulate the bipolar system and used to record EMGs from the MG muscle, are indicated with gray and black circles.

**Figure 3 sensors-24-03701-f003:**
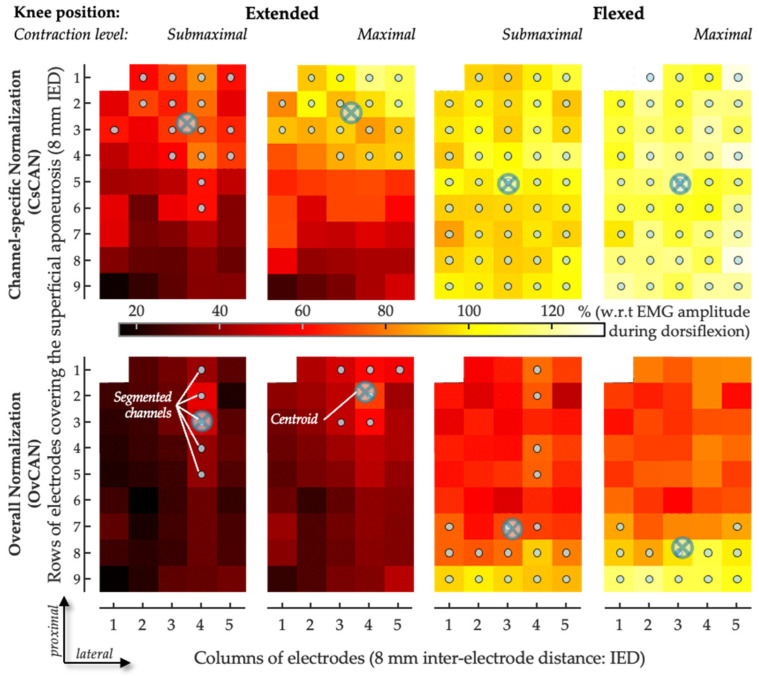
Sample of spatial distribution of coefficient of antagonist activation. The CAN values obtained for the medial gastrocnemius are represented as scaled images, with white and black pixels respectively denoting the highest and smallest CAN values. The top and bottom rows respectively show images for channel-specific and overall normalizations. For each channel, values were obtained over the whole 5-s recording. Gray circles indicate channels detecting the greatest CAN values, as identified with the automated segmentation procedure. The centroid coordinates along the transverse (columns) and longitudinal (rows) directions are represented with crossed circles.

**Figure 4 sensors-24-03701-f004:**
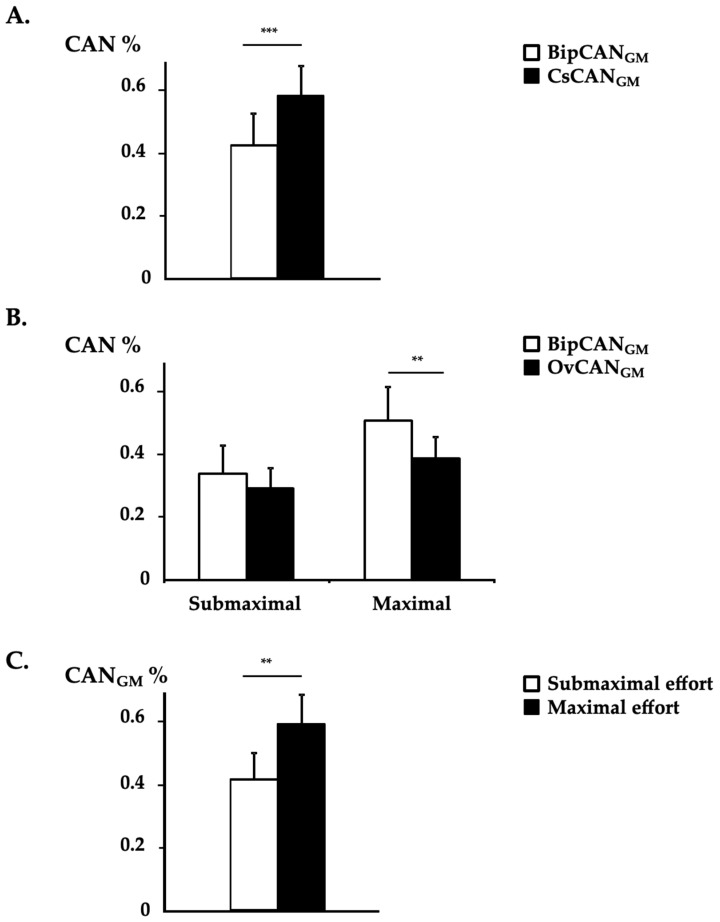
Coefficient of ANtagonist activation in the gastrocnemius medialis. The coefficients of agonist activation with bipolar EMG (BipCAN) and HD-EMG using channel-specific normalization (CsCAN) and overall normalization (OvCAN) from the gastrocnemius medialis (GM) in 12 patients (means ± standard errors of the means) assessed in a seated position. (**A**) Results of mean BipCANGM and CsCANGM with data from both effort levels and from both knee position conditions pooled. (**B**) Results of mean BipCANGM and OvCANGM by effort level with data from both knee position conditions pooled. (**C**) Results of mean CANGM in submaximal and maximal effort conditions with data from both EMG systems and from both knee position conditions pooled. **, *p* < 0.01; ***, *p* < 0.001.

**Figure 5 sensors-24-03701-f005:**
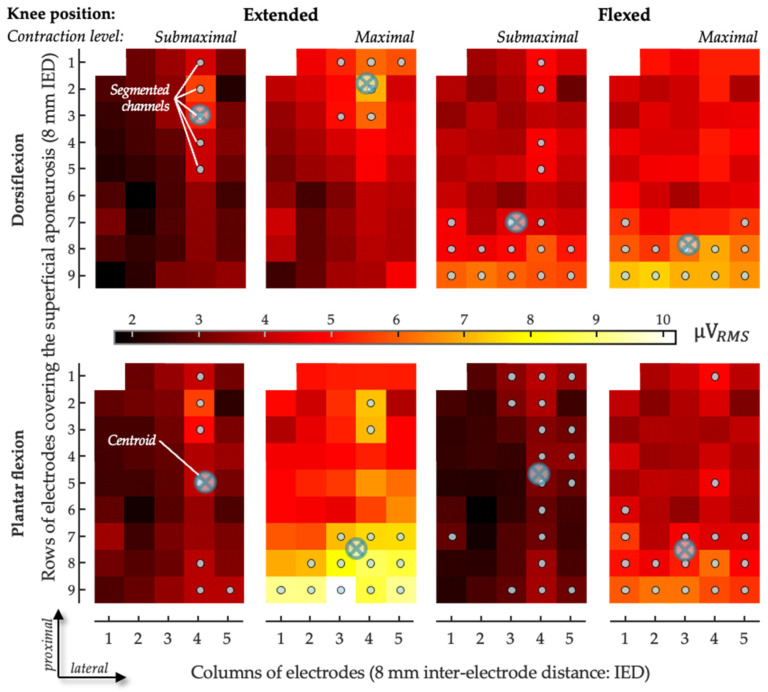
Sample of spatial distribution of coefficient of antagonist activation. The CAN values obtained for the gastrocnemius medialis are represented as scaled images, with white and black pixels respectively denoting the highest and smallest CAN values. The top and bottom rows respectively show images for dorsiflexion effort and plantar flexion effort recorded with the knee flexed and the knee extended in submaximal and maximal efforts. For each channel, values were obtained over the whole 5-s recording. Gray circles indicate channels detecting the greatest CAN values, as identified with the automated segmentation procedure. The centroid coordinates along the transverse (columns) and longitudinal (rows) directions are represented with crossed circles.

**Table 1 sensors-24-03701-t001:** Subject characteristics.

Subjects (n)	12
Age (y)	51 ± 14
Time since paresis onset (y)	6 ± 6
*Gender*	
Female (n)	3
Male (n)	9
*Paretic side*	
Left (n)	7
Right (n)	5
*Cause*	
Ischemic stroke (n)	9
Hemorrhagic stroke (n)	3

Values expressed as means ± SDs.

**Table 2 sensors-24-03701-t002:** Coefficients of antagonist activation.

Muscle	Knee Position	Effort Level	BipCAN	ReCAN	AbCAN
**Gastrocnemius medialis**					
n = 12	Flexed	Submaximal	0.33 ± 0.30	0.51 ± 0.33	0.30 ± 0.21
		Maximal	0.48 ± 0.42	0.71 ± 0.33	0.41 ± 0.23
	Extended	Submaximal	0.34 ± 0.33	0.47 ± 0.32	0.28 ± 0.23
		Maximal	0.53 ± 0.36	0.63 ± 0.33	0.36 ± 0.24
**Medial Soleus**					
n = 12	Flexed	Submaximal	-	0.40 ± 0.35	0.29 ± 0.23
		Maximal	-	0.48 ± 0.38	0.35 ± 0.25
	Extended	Submaximal	-	0.61 ± 0.76	0.54 ± 0.79
		Maximal	-	0.63 ± 0.43	0.49 ± 0.33
**Lateral Soleus**					
n = 12	Flexed	Submaximal	-	0.37 ± 0.32	0.27 ± 0.23
		Maximal	-	0.48 ± 0.35	0.34 ± 0.25
	Extended	Submaximal	-	0.41 ± 0.42	0.40 ± 0.59
		Maximal	-	0.61 ± 0.38	0.47 ± 0.31

Values expressed as means ± SDs of the coefficient of antagonist activation assessed with bipolar EMG (BipCAN) and high-density EMG using channel-specific (CsCAN) and overall (OvCAN) normalizations in two knee positions (flexed and extended) and at two effort levels (submaximal, 30% MVC, and maximal, 100% MVC). Abbreviations: BipCAN, coefficient of antagonist activation from bipolar EMG; CsCAN, coefficient of antagonist activation from HD-EMG with channel-specific normalization; and OvCAN, coefficient of antagonist activation from HD-EMG with overall normalization.

**Table 3 sensors-24-03701-t003:** Center of distribution of antagonist activation in GM.

	Knee Position	Effort Level	Absolute	Relative
**Location of the CAN centroid along proximal-distal axis**				
	Flexed	Submaximal	0.56 ± 0.16	0.49 ± 0.18
		Maximal	0.46 ± 0.13	0.42 ± 0.11
	Extended	Submaximal	0.54 ± 0.17	0.37 ± 0.16
		Maximal	0.64 ± 0.25	0.42 ± 0.19
**Location of the CAN centroid along medial-lateral axis**				
	Flexed	Submaximal	0.63 ± 0.17	0.68 ± 0.14
		Maximal	0.61 ± 0.14	0.64 ± 0.14
	Extended	Submaximal	0.67 ± 0.22	0.60 ± 0.15
		Maximal	0.60 ± 0.17	0.57 ± 0.16

Values expressed as means ± SDs of the normalized centroid location of the antagonist activation were quantified from the HD-EMG in the GM using channel-specific and overall normalizations in two knee positions (flexed and extended) and at two effort levels (submaximal, 30% MVC, and maximal, 100% MVC).

## Data Availability

Data are contained within the article.

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
