# Peer review of "Antagonist Activation Measurement in Triceps Surae Using High-Density and Bipolar Surface EMG in Chronic Hemiparesis"

_sensors, 2024, doi:10.3390/s24123701_

Round 1
Reviewer 1 Report
Comments and Suggestions for Authors
Overall, this draft makes a great contribution to the field, by exploring the effectiveness of two EMG technologies in measuring antagonist muscle activation in patients with chronic hemiparesis. I have attached a list of comments and suggestions below, hoping to make the draft more accessible. After minor modifications, the draft should be qualified to be published.

Author Response
Reviewer 1
Overall, this draft makes a great contribution to the field, by exploring the effectiveness of two EMG technologies in measuring antagonist muscle activation in patients with chronic hemiparesis. I have attached a list of comments and suggestions below, hoping to make the draft more accessible. After minor modifications, the draft should be qualified to be published.
1. Section 2.2, if possible, a simple schematic figure could be added to better illustrate the
experiment setup.
We thank the Reviewer for these comments. A figure illustrating the experimental setup is now added in the article. We presented surface electrodes monitoring EMG activity from ankle plantar flexors across two knee positions: flexed 90° and extended 0° at maximal isometric voluntary contraction (MVC) and then submaximal contractions at 30% MVC. An isometric brace with strain gauge measured dorsiflexor and plantar flexor torque exerted around the ankle.
2. Line 127, model and vendor of abrasive paste is suggested to be added here.
We thank the Reviewer for this comment. The model of the abrasive gel (Nuprep Skin Prup Gel, weaver and compagny) is now specified in the article.
3. Results Table 1, it is suggested to add some analysis on whether test subjects tend to endure
paresis around similar age, or the paresis happens completely randomly.
We thank the Reviewer for this comment. The reduced descending command observed in agonists (i.e. paresis) after stroke was well described in the literature. The exploration of potential correlations with the age should be indeed the subject of a specific questioning and study.
4. Line 411, regarding the second issue, when results cannot be genralized, what would be a
reasonable trend prediction?
We thank the Reviewer for this comment. Only the very distal fraction of the soleus muscle is accessible for surface electromyography (EMG) recordings. This is because the two proximal thirds of the soleus are covered by the gastrocnemius muscle (GM). The present analysis did not reveal any medio-lateral differences in CAN values for the soleus muscle. This suggests that antagonist activation is relatively uniform across the accessible portion of the soleus that was studied. The results from this analysis cannot be generalized to the entire volume of the soleus muscle. The recordings and findings are limited to the distal portion that is accessible via surface EMG. Surface EMG has inherent limitations when analyzing muscle activity at greater depths. This is particularly relevant for the soleus muscle, which has significant portions not accessible to surface EMG due to being covered by other muscles. While the analysis provides valuable insights into the degree of antagonist activation, these estimations should be indeed interpreted with caution. The findings may not accurately reflect the entire soleus muscle's behavior or activation patterns. We therefore feel that advancing any prediction would be too speculative for the moment: contrarily to biarticular leg muscles, for which proximo-distal variations in excitation have been consistently reported (Watanabe et al. 2012; Vieira et al. 2019), no proximo-distal differences in excitation within the soleus muscle have been documented.
5. In the conclusion part, the authors could elaborate a little more on future directions of
improvement for clinical procedures.
We thank the Reviewer for this comment. We now added this paragraph in the conclusion:
“HD-EMG could be used to develop personalized rehabilitation programs based on individual antagonist activations and neuromuscular capacities. Indeed, using HD-EMG could be the way to enhance patient education about the role and benefits of their active rehabilitation programs, improving adherence and engagement with their therapeutic protocols. Moreover, the usefulness efficacy of botulinum toxin injections used to treat antagonist activation has been suggested to depend on the toxin diffusion from the injection site. HD-EMG could be an efficient method to develop a functional model for predicting the number and location of muscles fibres paralysed due to the action of the botulinum toxin”.
References:
Watanabe K, Kouzaki M, Moritani T. Task-dependent spatial distribution of neural activation pattern in human rectus femoris muscle. J Electromyogr Kinesiol. 2012;22:251-8.doi: 10.1016/j.jelekin.2011.11.004.
Vieira TM, Lemos T, Oliveira LAS, Horsczaruk CHR, Freitas GR, Tovar-Moll F, Rodrigues EC. Postural Muscle Unit Plasticity in Stroke Survivors: Altered Distribution of Gastrocnemius' Action Potentials. Front Neurol. 2019;26;10:686. doi: 10.3389/fneur.2019.00686.

Reviewer 2 Report
Comments and Suggestions for Authors
This manuscript concerns the level of antagonist muscle activation in triceps surae using high-density and bipolar surface EMG in chronic hemiparesis.
This article is interesting, but I have a few doubts.
Line 60 – The authors refer to citation (28); -why they did not include an appropriate control group in this design;
-and did not measure the activity of the tibialis anterior muscle.
I ask the authors, to describe the limitations of the project more clearly.
1. What is the main question addressed by the research?
This manuscript concerns the level of antagonist activation in triceps surae using high-density, and bipolar surface EMG in chronic hemiparesis.
2. What parts do you consider original or relevant for the field? What specific gap in the field does the paper address?
An essential part concerns the methodology of measuring and calculating muscle activation, especially antagonists, using high-density and bipolar sEMG using an electrode grid, which is not very often used.
3. What does it add to the subject area compared with other published material?
Realisation of the objectives of the work, mainly examining the activity of antagonist muscles in chronic paresis patients, whot have not been assessed before.
4. What specific improvements should the authors consider regarding the methodology? What further controls should be considered?
I have no comments regarding the methodology. Future studies may consider tibialis anterior control activity, to control antagonistic activity better, and may also provide a more precise answer to the question of which technique can be used for specific clinical purposes.
5. Please describe how the conclusions are or are not consistent with the evidence and arguments presented. Please also indicate if all main questions posed were addressed and by which specific experiments.
The conclusions relate to the objectives and results of the research.
6. Are the references appropriate?
The references are appropriate.
7. Please include any additional comments on the tables and figures and quality of the data.
I have no comments regarding the tables and figures and the quality of the data.
Author Response
Reviewer 2
This manuscript concerns the level of antagonist muscle activation in triceps surae using high-density and bipolar surface EMG in chronic hemiparesis.
This article is interesting, but I have a few doubts.
Line 60 – The authors refer to citation (28); -why they did not include an appropriate control group in this design;
We thank the Reviewer for the useful comment. The present article concerned the comparison between the two systems in one population of patients. Indeed, a new study will compare hemipatetic subjects with healthy subjects using the HD-EMG system.
-and did not measure the activity of the tibialis anterior muscle.
We thank the Reviewer for this comment. The present work focused on the antagonist activation in the triceps surae. The reduced descending command observed in agonists (i.e. paresis) after stroke was well described in the literature. The quantification of the agonist activation of the tibialis anterior during dorsiflexion with HD-EMG should indeed be the subject of further study.
I ask the authors, to describe the limitations of the project more clearly.
We thank the Reviewer for these comments. The title of the paragraph 4.2. indicated more clearly the limitations of the study, which are detailed in this section of the discussion.
1. What is the main question addressed by the research?
This manuscript concerns the level of antagonist activation in triceps surae using high-density, and bipolar surface EMG in chronic hemiparesis.
2. What parts do you consider original or relevant for the field? What specific gap in the field does the paper address?
An essential part concerns the methodology of measuring and calculating muscle activation, especially antagonists, using high-density and bipolar sEMG using an electrode grid, which is not very often used.
3. What does it add to the subject area compared with other published material?
Realisation of the objectives of the work, mainly examining the activity of antagonist muscles in chronic paresis patients, whot have not been assessed before.
4. What specific improvements should the authors consider regarding the methodology? What further controls should be considered?
I have no comments regarding the methodology. Future studies may consider tibialis anterior control activity, to control antagonistic activity better, and may also provide a more precise answer to the question of which technique can be used for specific clinical purposes.
5. Please describe how the conclusions are or are not consistent with the evidence and arguments presented. Please also indicate if all main questions posed were addressed and by which specific experiments.
The conclusions relate to the objectives and results of the research.
6. Are the references appropriate?
The references are appropriate.
7. Please include any additional comments on the tables and figures and quality of the data.
I have no comments regarding the tables and figures and the quality of the data.

Reviewer 3 Report
Comments and Suggestions for Authors
The paper describes the difference in antagonist action in triceps surae during ankle dorsiflexion in chronic hemiparetic patients using two types of emg sensing systems. The effects of normalization on emg data analysis was also investigated.
The methods and results were presented in detail, and the discussion of results was thorough. The main observations to improve the presentation of this article are (1) to clarify the practical implications of the results in the abstract to match the stated problem and study objectives, and (2) to describe and demonstrate the practical implementation of the results for clinicians and/or clinical practice e.g. how the difference in sensor performance could assist the diagnosis, monitoring and/or treatment for chronic hemiparetic patients.
Author Response
Reviewer 3
The paper describes the difference in antagonist action in triceps surae during ankle dorsiflexion in chronic hemiparetic patients using two types of emg sensing systems. The effects of normalization on emg data analysis was also investigated.
The methods and results were presented in detail, and the discussion of results was thorough. The main observations to improve the presentation of this article are (1) to clarify the practical implications of the results in the abstract to match the stated problem and study objectives, and (2) to describe and demonstrate the practical implementation of the results for clinicians and/or clinical practice e.g. how the difference in sensor performance could assist the diagnosis, monitoring and/or treatment for chronic hemiparetic patients.
We thank the Reviewer for this comment.
We added this paragraph in the abstract:
“HD-EMG could be used to develop personalized rehabilitation programs based on individual antagonist activations and neuromuscular capacities”.
and this paragraph in the conclusion:
“HD-EMG could be used to develop personalized rehabilitation programs based on individual antagonist activations and neuromuscular capacities. Indeed, using HD-EMG could be the way to enhance patient education about the role and benefits of their active rehabilitation programs, improving adherence and engagement with their therapeutic protocols. Moreover, the usefulness efficacy of botulinum toxin injections used to treat antagonist activation has been suggested to depend on the toxin diffusion from the injection site. HD-EMG could be an efficient method to develop a functional model for predicting the number and location of muscles fibres paralysed due to the action of the botulinum toxin”.
